# Revised Harris–Benedict Equation: New Human Resting Metabolic Rate Equation

**DOI:** 10.3390/metabo13020189

**Published:** 2023-01-28

**Authors:** Eleni Pavlidou, Sousana K. Papadopoulou, Kyriakos Seroglou, Constantinos Giaginis

**Affiliations:** 1Department of Food Science and Nutrition, School of Environment, University of the Aegean, 81400 Myrina, Lemnos, Greece; 2Department of Nutritional Sciences and Dietetics, International Hellenic University, 57400 Thessaloniki, Greece

**Keywords:** resting energy expenditure (REE), basal metabolic rate (BMR), resting metabolic rate (RMR), indirect calorimetry (IC), body mass index (BMI), predictive equations, metabolism, weight management, obesity, overweight

## Abstract

This paper contains a revision of the Harris–Benedict equations through the development and validation of new equations for the estimation of resting metabolic rate (RMR) in normal, overweight, and obese adult subjects, taking into account the same anthropometric parameters. A total of 722 adult Caucasian subjects were enrolled in this analysis. After taking a detailed medical history, the study enrolled non-hospitalized subjects with medically and nutritionally controlled diseases such as diabetes mellitus, cardiovascular disease, and thyroid disease, excluding subjects with active infections and pregnant or lactating women. Measurement of somatometric characteristics and indirect calorimetry were performed. The values obtained from RMR measurement were compared with the values of the new equations and the Harris–Benedict, Mifflin–St Jeor, FAO/WHO/UNU, and Owen equations. New predictive RMR equations were developed using age, body weight, height, and sex parameters. RMR males: (9.65 × weight in kg) + (573 × height in m) − (5.08 × age in years) + 260; RMR females: (7.38 × weight in kg) + (607 × height in m) − (2.31 × age in years) + 43; RMR males: (4.38 × weight in pounds) + (14.55 × height in inches) − (5.08 × age in years) + 260; RMR females: (3.35 × weight in pounds) + (15.42 × height in inches) − (2.31 × age in years) + 43. The accuracy of the new equations was tested in the test group in both groups, in accordance with the resting metabolic rate measurements. The new equations showed more accurate results than the other equations, with the equation for men (R-squared: 0.95) showing better prediction than the equation for women (R-squared: 0.86). The new equations showed good accuracy at both group and individual levels, and better reliability compared to other equations using the same anthropometric variables as predictors of RMR. The new equations were created under modern obesogenic conditions, and do not exclude individuals with regulated (dietary or pharmacological) Westernized diseases (e.g., cardiovascular disease, diabetes, and thyroid disease).

## 1. Introduction

Metabolism refers to the set of biochemical reactions that take place in the cells of living organisms. Basic metabolism includes all the biochemical processes in the organism involved in the production and release of energy to sustain life during a period of complete rest [1].

Basal metabolic rate (BMR) represents the amount of energy in kilocalories used over a given period of time (e.g., 24 h) to perform the most basic functions of the body. BMR can be accurately assigned under very restrictive conditions of a laboratory or hospital environment: thermally neutral, in the post-absorptive stage, after sleep, in an awake state, during complete rest, during physical and mental calm, and avoiding stimulation of the sympathetic nervous system. It can be measured by direct calorimetry (DC) and more often by indirect calorimetry (IC). BMR is the lowest metabolic rate after the sleep metabolic rate. It represents 60 to 75% of daily calorie expenditure for most people and decreases after the second decade of life by 1–2% per decade, mainly due to changes in metabolically active muscle tissue [2].

Resting metabolic rate (RMR) represents the amount of energy the body needs to function while at rest. The measurement to determine RMR differs from the measurement of BMR in that it does not need to be performed before getting out of bed, as it involves low-effort daily activities such as using the bathroom, dressing, gentle movement, etc., in addition to basic body functions. It is usually measured by indirect calorimetry, in the morning before the first meal and after abstinence from factors that may affect the metabolic rate such as exercise, caffeine and alcohol consumption, and smoking. In the modern sedentary lifestyle, RMR accounts for most of the energy expended during the day. RMR is slightly higher by about 10% than BMR due to the contribution of low energy expenditure [3].

Resting energy expenditure (REE) represents the energy expended by humans in an awake, resting, interstitial state. It is considered by the American Research Council to be an equivalent term to RMR and is used to calculate resting energy processes [4]. In the international literature, the terms BMR, RMR, and REE are usually confused and applied for exactly the same purpose. Thus, the term REE is used by the Harris–Benedict [5,6], Roza and Shizgal [7], and Mifflin–St Jeor [8] equations; the term BMR by the FAO/WHO/UNU equations [9]; and the term RMR by the OWEN equations [10].

Both direct and indirect calorimetry measurements that can provide an accurate measurement of metabolic rate are expensive and require trained personnel, special site conditions, and subject preparation. In this aspect, mathematical formulas have been developed for the indirect determination of RMR, usually based on parameters of healthy adults, such as body weight, height, age, and sex [5,6,7,8,9,10,11,12]. In recent years, in the context of adapting prediction equations to specialized categories of individuals and situations, several prediction equations have been developed that are specific to particular diseases [13,14,15,16,17], ethnicity [18,19], special categories such as athletes [20,21], overweight and obese people [22,23], children [24], and people after middle age [25]. Several attempts to create new RMR prediction equations using individual characteristics, such as free fat mass (FFM), have emerged from time to time but they require the use of body composition analysis devices [26]. Some of the widely used prediction equations are listed below.

### 1.1. REE, BMR, and RMR Predictive Equations

#### 1.1.1. Harris–Benedict

Based on the literature, the Harris–Benedict (H–B) equations [5,6] constitute the principle of creating the equations (Harris–Benedict principle) without the help of modern computers. They were published more than 100 years ago (1918 and 1919) and remain still the most frequently used equations in daily practice. These equations were based on 239 subjects (136 men and 108 women), aged 16–63 years (Table 1). The equation for males had a coefficient of R^2^ = 0.64 and the equation for females had a sample correlation coefficient of R^2^ = 0.36.

#### 1.1.2. Roza and Shizgal

The first revision of the H–B equations was carried out 65 years later by Roza and Shizgal (1984), based on a sample that was older, larger by 98 persons (*n* = 337), and almost equally divided between the sexes (168 men and 169 women). These equations (Table 1) showed stronger correlation coefficients (R^2^ = 0.77 and R^2^ = 0.68 for men and women, respectively) for both sexes than the H–B equations [7].

#### 1.1.3. Mifflin–St Jeor

The second revision/simplification of the H–B equations was carried out 71 years later (1990) by Mifflin–StJeor, based on a sample that was wider in age (19–78 years) and larger by 259 individuals (*n* = 498), of which 251 were men and 247 were women (Table 1). These equations showed a common coefficient R^2^ = 0.71 for both males and females [8].

#### 1.1.4. FAO/WHO/UNU Equations

The Food and Agriculture Organization/World Health Organization/United Nations University (FAO/WHO/UNU) equations were developed in 1985 (Table 1) using equations derived mainly from studies in Western Europe and North America [12]; almost half of the data came from studies between the late 1930s and early 1940s involving Italian men with relatively high BMR values. These equations were based on a large population sample (*n* = 11000) and categorized into six age groups (<3, 3–10, 10–18, 18–30, 30–60, and >60 years), with R^2^ ranging between 0.60–0.97 for men and 0.70–0.97 for women [9].

#### 1.1.5. Owen

Owen, in two consecutive years, created new, simpler equations (Table 1) using body weight as the only parameter. In 1986, he presented the equation for women with a coefficient r = 0.74, which was based on a sample of 44 women aged 18–65 y whose body weight ranged between 43 and 171 kg [10]. A year later (1987), he presented the equation for men with a coefficient similar (R^2^ = 0.56) to that for women (R^2^ = 0.54), which was based on a sample of 60 men aged 18–82 years and weighing 60–171 kg [11].

The aforementioned equations, and other more specific equations used extensively for body weight management, have been evaluated for their validity and reliability by a plethora of studies, which indicates a tendency to overestimate or underestimate REE compared with indirect calorimetry [27,28].

The changes that have occurred in people’s lifestyles, diets, work, and physical activity since the first equations were created have resulted in the diversification of somatometric characteristics due to a considerable increase in overweight/obesity, which, based on World Health Organization data, has almost tripled between 1975 and 2016.

The global prevalence of obesity between 1975 and 2016 has almost tripled [29] and the proportion of obesity-related deaths from 1990 to 2017 has increased by 8% [30]. This upward trend in the global obesity rate will continue, as according to the fourth World Atlas of Obesity, between 2010, 2025, and 2030 it will rise from11.4% to 16.1% to 17.5%, respectively [31]. Studies from various European countries support this steady increase in the prevalence of overweight/obesity, which they attribute to a combination of unhealthy diets and physical inactivity, unhealthy body weight in early life, environmental factors, digital marketing of unhealthy foods to children, sedentary lifestyles, online gaming, and to other factors [32] that did not exist during the time period when the H–B equations were created.

Therefore, the aim of this study was to revise the Harris–Benedict equation, 104 years after its original version and 32 years after its last revision by Mifflin–St Jeor, using the same easily measurable anthropometric indicators (weight, height, age, and sex).

## 2. Materials and Methods

### 2.1. Participants

A total of 722 Caucasian subjects (173 men and 549 women) were included in this retrospective study after receiving their full medical history. Non-hospitalized individuals with common diseases such as cardiometabolic diseases (e.g., hypertension, coronary artery disease, prediabetes, diabetes mellitus, hyperlipidemias, thyroid diseases, etc.), which are controlled by either dietary or medication (e.g., antidiabetic, anti-lipidemic, benzodiazepines, etc.), were not excluded from the sample. Subjects with pregnancy [33], lactation [34], active infections [35], and unregulated diseases were excluded from participation in the study. The characteristics of the participants are presented in Table 2.

### 2.2. Predictive Equations

The H–B, Mifflin–St Jeor, FAO/WHO/UNU, and Owen equations were used to compare with the new equations.

### 2.3. Measures

#### 2.3.1. Anthropometric Measurements

The Tanita device (MC-780, Tanita Corporation, Tokyo, Japan) was used to measure body weight with an accuracy of 0.1 kg. Measurement was performed according to protocol with light clothing, without shoes, socks, or stockings, and with arms and legs slightly separated from the body. All measurements were performed during the same visit and in the early morning hours.

A Seca 222 wall-mounted stadiometer with an accuracy of 0.5 cm was used to measure stature utilizing a standard technique, without shoes and hair ornaments, with the legs straight, joined and resting on the wall, arms at the side, shoulders flat, the angle of view parallel to the floor, head, shoulders, buttocks, and heels in contact with the flat surface (wall) [36].

Body mass index was calculated by dividing weight (in kilograms) by the square of height (in meters).

#### 2.3.2. Indirect Calorimetry

The IC device, Cosmed Fitmate Pro, which has demonstrated test/retest/retest reliability both within a day and between two different days, was used to measure RMR [37,38]. This device measures the Respiratory Quotient (RQ), i.e., the ratio of carbon dioxide (CO2) produced by the body to oxygen (O2) consumed by the body. The tests were performed with a face mask.

The procedure was performed using the best-appropriate practice methods [39].

Participants were instructed to avoid vigorous activity for 12 h before the visit and to abstain from food, energy drinks, coffee, alcohol, and nicotine for at least 8 h. The examination was performed in a thermally neutral room, in a quiet environment, and after 10 min of rest. The same procedure was repeated one week later for each participant and the mean RMR value was used.

For statistical convenience and to distinguish between equation-predicted RMR values, and IC-measured RMR values, RMRP and RMRIC were assigned, respectively.

### 2.4. Statistical Analysis

The statistical analysis was performed mainly in statistical package R (R version 4.1.3 (2022). In the creation of the formula, the initial approaches used different types of regressions (linear, polynomial, logarithmic) and linear regression had the most accurate results based on the coefficient of determination (R^2^) and standard error of the estimate (SEE). The Kolmogorov–Smirnov test and the Shapiro–Wilk test were used to examine whether variables were normally distributed (the subjects were randomly assigned to the training or test subset in such a way that the ratio between sexes remained constant (for males 133 and 40 and for females 429 and 127). To minimize bias and to optimize the validation data, k-Fold cross-validation was applied with linear regression for the production of the equations for female and male training groups. In R to apply k-Fold cross-validation, we used the caret package (version 6.0-93, Max Kuhn, can be found on CRAN and the project is hosted on GitHub). In summary, what k-Fold cross-validation does is shuffle the training dataset randomly and split it into equal groups (if possible), then, for each group, the following procedure is followed: (1) Take the group as a holdout or validation data set; (2) Take the remaining groups as a training data set; (3) Fit a model on the training set and evaluate it on the holdout set; (4) Retain the evaluation score and discard the model. After that procedure, it summarizes the skill of the model using the sample of model evaluation scores. Each observation in the data sample is assigned to an individual group and stays in that group for the duration of the procedure. Thus, after the observations split into groups they remain in the same group during the whole procedure.

## 3. Results

The final results showed that for both sexes the weight was significantly associated with RMR_IC_ (for both genders *p*-value < 0.001). On the other hand, regarding height, the results suggest that it is more significant for females (*p*-value = 0.001) than for males (*p*-value = 0.065). Lastly, the results indicated that the age of both sexes is significant but slightly more for males (*p*-value = 0.005) than females (*p*-value = 0.011). It should be noted that the resampling results gave us more accurate results for males (R-squared: 0.95) than females (R-squared: 0.86). The new equations for both sexes in both the metric and imperial systems are presented in Table 1 and below:

### 3.1. International System of Units

RMR males = (9.65 × weight in kg) + (573 × height in m) − (5.08 × age in years) + 260

RMR females = (7.38 × weight in kg) + (607 × height in m) − (2.31 × age in years) + 43

### 3.2. Imperial System

RMR males = (4.38 × weight in pounds) + (14.55 × height in inches) − (5.08 ×age in years) + 260

RMR females = (3.35 × weight in pounds) + (15.42 × height in inches) − (2.31 ×age in years) + 43

### 3.3. Differences between Predictive Equations and Measurement RMR

The difference between the predictive equations (RMR_P_) and measurement RMR_IC_, the percent bias (accuracy at the group level), and the root mean square error for both the new equations and selected equations are described in Table 3 and Table 4. We found that both the male equation and the female equation showed a bias of <9%. Specifically, the mean bias of the male equation was 8.3% and that of the female equation was 8.9%. Among the selected equations from the literature, we found that the introduced equation in both men and women was the most accurate at the group level (mean bias 8.32 and 8.93, respectively). Table 5 and Table 6 illustrates some more metrics of the differences between RMR_P_ and RMR_IC_ that validate that the most successful prediction for males and females comes from the introduced equation and the H–B equation, respectively.

Regarding the accuracy at the individual level, the percentages of participants with an RMR_P_ within ±10% of the RMR_IC_ for the new and other predictive equations are reported in Figure 1. The new equation reported the highest accuracy in men together with Harris–Benedict (67.5% and 65%, respectively), and the same two equations reported the highest accuracy in women (59.1% and 57.5%, respectively) when compared to the other equations. It should be noted that the least-accurate equations tend to under predict RMR_P_ in both sexes when not accurate.

### 3.4. Bland–Altman Plots of RMR_P_-RMR_IC_ Differences

The Bland–Altman plots of predicted–measured RMR differences vs. mean predicted–measured RMR obtained by all equations are depicted in Figure 2. There is a good agreement for both the H–B equation and the new equations.

## 4. Discussion

The results of this present study are important because they were conducted under the conditions of the modern “fat-inducing” environment of prolonged sedentary behavior, low levels of physical activity, abundant consumption of energy-rich, over-processed foods, etc., which partly explains the increased mean BMI; this slightly drags up energy needs, compared to the H–B equation. In the second decade of the 1900s, when the H–B equations were created, the average REE was about 1400 to 1600 calories per day for women and men, respectively [40]. According to our study, in the modern sedentary lifestyle of the 2020s, the average RMR ranges between 1500 kcal for women and 2000 kcal for men. Other studies also report a much higher average RMR of 3000 calories/day, mainly for men [41]. This variation in metabolic rate creates the need for revisions of the original H–B equations.

Another important point of this present study relates to the fact that the new proposed equations were based on a broader body mass index (BMI: 17–48 kg/m^2^) population sample than the original H–B equations (BMI: 12.3–32.5 kg/m^2^), in which the proportion of subjects with BMI > 30 kg/m^2^ was only 5% [5,6].

A particularly important point to note about the new equations proposed by our study is the fact that the population sample included both healthy individuals who were not receiving medication and individuals with high-prevalence diseases that were controlled nutritionally and/or pharmacologically. This fact facilitates the use of these equations in daily practice, as it does not preclude their use in population groups associated with diseases with increased rates [42,43], constituting an important part of the population in need of nutritional support.

An interesting aspect of the new equations is that they showed small reductions in prediction error and overestimation bias compared with the measured RMR_IC_ values. The fact that all equations manifest strong reliability in the population from which the data of [44] were drawn is counterbalanced by the better response shown by the new equations compared to other equations presented in this study (H–B, Mifflin–St Jeor, FAO/WHO/UNU, and Owen).

The importance of body weight in improving the predictability of the equations for both sheets, observed in this present study, strengthens the data supporting the contribution of body size to metabolic rate [45,46].

The involvement of age as an important factor in the predictability of the equations presented by the current study has been supported by other studies [47], which is probably based on the modifications of body composition (muscle tissue and lean mass) over time.

However, there are also limitations in our study, which are related to the complete absence of the underweight category from the sample of men (20 to 48 kg/m^2^) and of individuals overall with BMI < 17 kg/m^2^.

In this present study, better coefficients for linear relationships were verified, however, allometric relationships have not been considered, which have been shown by other studies to increase the reliability of predicting both BMR and RMR in healthy subjects, adding another level of accuracy to calculations [44].

Another limitation of this present study that should be pointed out concerns the lower population sample of men compared with that of women, but this is not a particularly important drawback as it is not smaller than that used in both the H–B and Roza and Shizgal equations [5,6,7]. In addition, the newly proposed predictive equations of men showed higher reliability than those of women.

## 5. Conclusions

In conclusion, the new resting metabolic rate prediction equations proposed by this study are reliable, easy to use, and can be widely used for body weight management when the measurement of RMR by direct and indirect calorimetry systems is not feasible. However, it is strongly recommended to derive new resting metabolic rate prediction equations for different population races beyond the Caucasian race with different genetic backgrounds, lifestyle factors, anthropometric characteristics, and nutritional habits.

## Figures and Tables

**Figure 1 metabolites-13-00189-f001:**
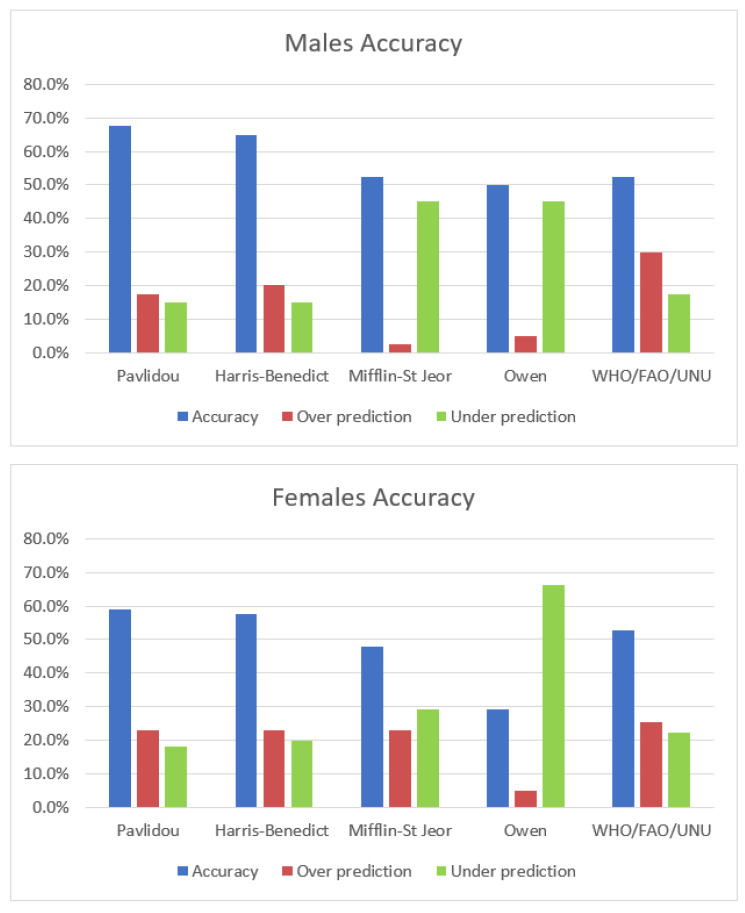
Prediction accuracy within ±10%. Accuracy of prediction equations for measurements of resting metabolic rate within ±10% using each equation in 40 men and 127 women, respectively.

**Figure 2 metabolites-13-00189-f002:**
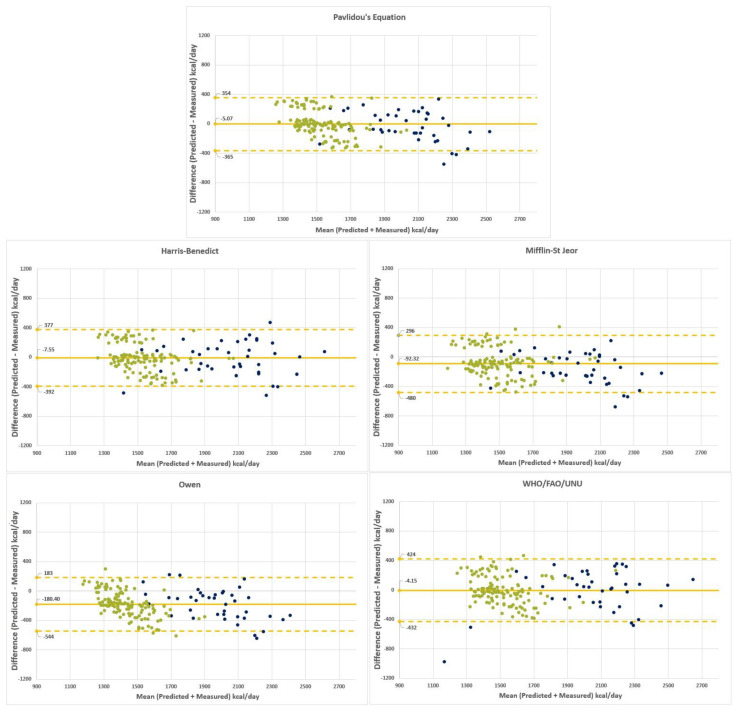
Bland–Altman plots. Bland–Altman plots display the mean of the differences between predicted and measured resting metabolic rates using the selected equations for men and women with a solid line. The dotted lines represent 1.96 SDs from the mean (limits of agreement). Note that the green dots are for females and the blue ones for males.

**Table 1 metabolites-13-00189-t001:** Equations for Human Resting Metabolic Rate.

Parameters	Predictive Equations	Population Description	Age	R^2^ Male	R^2^ Female	Male	Female
Wt, Gender, and Age Groups	FAO/WHO/UNU in kcal/d(1985)	*n* = 11,000 Many Ethnic groups and a broad BMI range	<3	0.97	0.97	(60.9 × wt in kg) − 54	(61. 3 × wt in kg) − 51
			3–10	0.86	0.85	(22.7 × wt in kg) + 495	(22.43 × wt in kg) + 499
			10–18	0.90	0.75	(17.5 × wt in kg) + 651	(12.2 × wt in kg) + 746
			18–30	0.65	0.72	(15.3 × wt in kg) + 679	(14.7 × wt in kg) + 496
			30–60	0.60	0.70	(11.6 × wt in kg) + 879	(8.7 × wt in kg) + 829
			>60	0.79	0.74	(13.5 × wt in kg) + 487	(10.5 × wt in kg) + 596
Wt and Gender	Owen in kcal/d (1986 and 1987)	*n* = 60 M, multiple racial/ethnic volunteers 18–82 y, 60–17 1 kg *n* = 44 F (included 8 athletes), no specific racial/ethnic information provided, 18–65 y, 43–143 kg.	adults	0.56	0.54	879 + (10.2 × wt in kg)	795 + (7.18 × wt in kg)
Wt, Ht, Age, Gender	Harris–Benedict in kcal/d (1918, 1919)	*n* = 239, White normalweight, 16–63 y, 136 males (M) (weight mean 61.1 ± 10.3 Kg and mean ages 27 ± 9 y) 103 females (F) (mean weight 56.5 ± 11.5 Kg and mean ages 31 ± 4 y), Over a ten-year period	adults	0.64	0.36	(13.75 × wt in kg) + (5.003 × ht in cm) − (6.755 × age in y) + 66.47	(9.563 × wt in kg) + (1.850 × htin cm) − (4.676 × age in y) + 655.1
	H–B rev. by Rosa and Shizgalin kcal/d (1984)	*n* = 337, 168 M, 169 F (with a wider age range from original H–B)	adults	0.77	0.68	(13.397 × wt in kg) + (4.799 × htin cm) − (5.677 × age in y) + 88.362	(9.247 × wt in kg) + (3.098 × htin cm) − (4330 × age in y) + 447.593
	Mifflin in kcal/d (1990)	*n* = 498, 19–78 y (mean ages 44 ± 14 y), 251 M (mean weight 87.5 ±14.4 Kg) 247 F (mean weight 70.2 ± 14.1 Kg)	adults	0.71	0.71	(9.99 × wt in kg) + (6.25 × ht in cm) − (4.92 × age in y) + 5	(9.99 × wt in kg) + (6.25 ×ht in cm) − (4.92 × age in y) − 161
	Pavlidou (Proposed New Equations), in kcal/d (2022)	*n* = 722, Caucasians 173 M, 18–78 y, 55–177 kg, BMI: 20–48 Kg/m^2^ *n* = 549 F, 19–76 y, 43–139 kg, BMI:17–47 Kg/m^2^	adults	0.95	0.86	(9.65 × wt in kg) + (573 × ht in m) − (5.08 × age in y) + 260	(7.38 × wt in kg) + (607 × ht in m) − (2.31 × age in y) + 43
						(4.38 × wt in pounds) + (14.55 × ht in inches) − (5.08 × age in y) + 260	3.35 × wt in pounds) + (15.42 × ht in inches) − (2.31 × age in y) + 43

**Table 2 metabolites-13-00189-t002:** Characteristics of the study population.

Characteristics	Males	Females	Total
*n*	173	549	722
Age range (years)	18 to 78	19 to 76	18 to 78
Age mean ± SD (years)	39 ± 13	38 ± 12	38 ± 13
Weight range (kg)	55 to 157	43 to 139	43 to 157
Weight mean ± SD (kg)	99 ± 19	78 ± 16	83 ± 19
Height range (m)	1.5 to 2.03	1.48 to 1.86	1.48 to 2.03
Height mean ± SD (m)	1.78 ± 0.08	1.64 ± 0.06	1.68 ± 0.09
BMI (kg/m^2^)	20 to 48	17 to 47	17 to 48
BMI mean ± SD (kg/m^2^)	31 ± 6	29 ± 6	29 ± 6
RMR_IC_ range (kcal/24 h)	1039 to 2595	908 to 2492	908 to 2595
RMR_IC_ mean ± SD (kcal/24 h)	2006 ± 346	1533 ± 308	1646 ± 376

**Table 3 metabolites-13-00189-t003:** Evaluation of selected predictive equations in 40 men based on differences Predicted (RMR_P_)-Measured (RMR_IC_), percentage of accuracy, Bias, and Root Mean Square Error.

Males	Difference RMR_P_-RMR_IC_Min (Kcal/d)	Difference RMR_P_-RMR_IC_Max (Kcal/d)	Bias % *	Max Negative Error	Max Positive Error	RMSE (Kcal/d)
Pavlidou (New Equation)	20.38	545.49	8.32%	−21.26%	16.39%	204.61
Harris–Benedict	8.51	513.7	9.06%	−28.83%	23.18%	222.71
Mifflin–StJeor	5	671	9.24%	−26.61%	11.12%	254.60
Owen	1.84	641.2	10.17%	−25.34%	14.35%	275.27
WHO/FAO/UNU	11.05	973.6	11%	−58.90%	21.31%	281.32

RMR_IC_, measured resting metabolic rate; RMR_P_, predicted resting metabolic rate; RMR, resting metabolic rate; RMSE, root mean square error; * mean percentage error between predictive equations and measured value.

**Table 4 metabolites-13-00189-t004:** Evaluation of selected predictive equations in 127 women based on differences Predicted (RMR_P_)-Measured (RMR_IC_), percentage of accuracy, Bias, and Root Mean Square Error.

Females	Difference RMR_P_-RMR_IC_Min (Kcal/d)	Difference RMR_P_-RMR_IC_Max (Kcal/d)	Bias % *	Max Negative Error	Max Positive Error	RMSE (Kcal/d)
Pavlidou (New Equation)	2.31	366.8	8.93%	−18.82%	27.2%	175.55
Harris–Benedict	1.76	378.65	9.49%	−20.62%	30.15%	186.21
Mifflin–StJeor	0.5	476	11.23%	−25.94%	26.9%	204.92
Owen	11.6	612.03	13.38%	−30.52%	26.01%	252.78
WHO/FAO/UNU	5.22	466.77	10.31%	−20.86%	38.88%	193.41

RMR_IC_, measured resting metabolic rate; RMR_P_, predicted resting metabolic rate; RMR, resting metabolic rate; RMSE, root mean square error; * mean percentage error between predictive equations and measured value.

**Table 5 metabolites-13-00189-t005:** Difference RMR_P_-RMR_IC_ in Males.

Males	Pavlidou (New Equation)	Harris–Benedict	Mifflin–StJeor	Owen	WHO/FAO/UNU
Average	−29.3	−12.17	−154.4	−178.28	−5.68
Average ABS	171.32	185.22	198.34	219.31	216.91
SD	205.08	225.21	205.02	212.42	284.85

Average ABS: the average of the absolute differences RMR_P_-RMR_IC_; SD: standard deviation.

**Table 6 metabolites-13-00189-t006:** Difference RMR_P_-RMR_IC_ in Females.

Females	Pavlidou (New Equation)	Harris–Benedict	Mifflin–StJeor	Owen	WHO/FAO/UNU
Average	2.42	−6.09	−72.77	−181.06	−3.66
Average ABS	131.78	140.47	170.4	213.54	153.8
SD	176.23	186.85	192.32	177.12	194.14

Average ABS: the average of the absolute differences RMR_P_-RMR_IC_; SD: standard deviation.

## Data Availability

The data that support the findings of this study are available from the corresponding author upon reasonable request due to privacy or ethical restrictions.

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
