# Peer review of "Revised Harris–Benedict Equation: New Human Resting Metabolic Rate Equation"

_metabolites, 2023, doi:10.3390/metabo13020189_

Round 1

Reviewer 1 Report

Dear Editor,

thanks so much for the opportunity to revise the work entitled "Revised Harris-Benedict Equation: New Human Resting Metabolic Rate Equation”.

The work is very interesting, showing the development and validation of new equations for the estimation of Resting Metabolic Rate (RMR) in normal, overweight and obese adult subjects, taking into account the same anthropometric parameters.

The paper is well written, the results clearly reported and the statistical methods rigorous.

I have not specific revisions for the authors to perform. I only suggest improving the discussion.

Thanks.

Author Response

Dear reviewer,

Thank you for your feedback there have been some changes to the discussion section. Hopefully, it will be better now.

Happy new year.

Kind regards.

Reviewer 2 Report

 The manuscript Revised Harris-BenedictEquation: New Human Resting Metabolic Rate Equation by Eleni Pavlidou,Sousana K Papadopoulou and Kyriakos Seroglou and Constantinos Giaginis is an attempt to refresh the equations allowing to calculate RMR in normal, overweight, and obese individuals using the same antropometric parameters as Harris-Benedict, Mifflin-St Jeor etc, i.e., age, body we4ight, height, and sex. 722 adult Greek participants were involved. RMR obtained with indirect calorimetry was considered a gold standard.

The new Pavlidou equations are provided in the graphical abstract, both for the SI and imperial systems.

They are characterized by the best accuracy of predicting the same(?) data, the model was developed on??? Do I get it correctly? This is not the way a model is created. The data should be divided into two groups. One of them should be used to develop the model, whereas the other to evaluate it.

Introduction provides an in-depth review of the theory behind the equation. All the currently used models are thoroughly described including the population, which they were developed for.

I believe, the manuscript will benefit from clear statement, why a new set of equations are needed (changing body composition in Europe etc). It would be preferable to provide this paragraph at the very beginning of the introduction. Currently, it is at the beginning of the discussion.

Were participants with controlled disease also included in developing the aforementioned models? If not, please highlight it in the abstract. This fact may lead to better generalizability of your equations, as compared to the earlier attempts. BMI and height distributions in this study also resemble those in the general population.

Please provide the numbers for test/retest reliability of the IC device in the manuscript. (paragraph 2.3.2)

Wouldn’t it be helpful to consider some parameters (FFM, body fat, etc) as other predictors in your equation? BI devices are available at gyms, therefore they could be obtained easily at least by a significant fraction of the population. If the BI-derived params will be described in another manuscript, do not mention them in this one. Unless you want to use them to better characterize your population, which is not the case.

Line 240 – remove “to note”

Please check the manuscript for spelling (accidently merged words) and punctuation (e.g, missing periods).

Author Response

Dear reviewer,

Thank you for your insightful comments on our paper. We have been able to
incorporate changes to reflect most of the suggestions provided by the reviewers. please find below the replies to your inputs. 

1)
Q: The new Pavlidou equations are provided in the graphical abstract, both for the SI and imperial systems.

A: The measurements on the subjects were retrieved using the SI system we converted the equations to the imperial system also for the readers’ convenience.

2)
Q: They are characterized by the best accuracy of predicting the same(?) data, the model was developed on??? Do I get it correctly? This is not the way a model is created. The data should be divided into two groups. One of them should be used to develop the model, whereas the other to evaluate it.

A: You are right the default procedure is the one you have mentioned. We indeed split the initial data into two groups and then we run the K fold cross-validation. In the paragraph where the process is explained validation, test, and training groups are mentioned but we did not how the split was done. We have added the information. On the comment on why we present all the subjects, it is quite common to use K-fold cross-validation without having a test group when there is limited data as long as we keep in mind and check the bias of the results [1]. In order to try to give a short explanation of why this can happen, it is because of how the K fold cross-validation works (with different subjects as validation group for each of the K models), and when having limited data it can be considered more accurate to include all the data in the test group. In more detail, in our case, there is an argument, especially for males, that since we did a 77-23 split (the most common is 80-20) it gave us 40 subjects, and taking into consideration that we cover 5 classes (normal weight, overweight, obesity class I, obesity class II, and obesity class III) it gives an average of 8 subjects per class in our test group and the performance can be challenged as an output might be based on the luck of the test group. In the end, what we did was to include all the data in the test group to minimize the luck of the test selection while using only 77 percent to run the K-fold cross-validation in order to not have included all the test data in the training process.

So to follow your guidelines we will provide the results of only the 23 percent that was left out, this is expected to have lower bias and at the same time make the results more vulnerable to the randomness of the selection of the test group.

[1] Vanwinckelen, G., Blockeel, H., De Baets, B., Manderick, B., Rademaker, M., & Waegeman, W. (2012). On estimating model accuracy with repeated cross-validation. In BeneLearn 2012: Proceedings of the 21st Belgian-Dutch Conference on Machine Learning. pp. 39

3)
Q: I believe, the manuscript will benefit from clear statement, why a new set of equations are needed (changing body composition in Europe etc). It would be preferable to provide this paragraph at the very beginning of the introduction. Currently, it is at the beginning of the discussion.

A: It has been included in the introduction

4)
Q: Were participants with controlled disease also included in developing the aforementioned models? If not, please highlight it in the abstract. This fact may lead to better generalizability of your equations, as compared to the earlier attempts. BMI and height distributions in this study also resemble those in the general population.

A: It has been included in the abstract

5)
Q: Please provide the numbers for test/retest reliability of the IC device in the manuscript. (paragraph 2.3.2)

A: It has been provided

6)
Q: Wouldn’t it be helpful to consider some parameters (FFM, body fat, etc) as other predictors in your equation? BI devices are available at gyms, therefore they could be obtained easily at least by a significant fraction of the population. If the BI-derived params will be described in another manuscript, do not mention them in this one. Unless you want to use them to better characterize your population, which is not the case.

A: It has been corrected

7)
Q: Line 240 – remove “to note”

A: It has been removed

8)
Q: Please check the manuscript for spelling (accidently merged words) and punctuation (e.g, missing periods).

A: Have been corrected.

Happy new year.

Kind regards.

Reviewer 3 Report

General questions

Congratulations on the work. However, some issues need to be clarified in addition to other small corrections (The text requires a thorough and detailed revision. There are extra spaces and missing spaces between words, abbreviations and numbers..)

Major questions

- The text (including tables) requires a thorough and detailed revision. There are extra spaces and missing spaces between words, abbreviations and numbers..

- Table 3: Please correct: very misconfigured

- Table 3: Please, insert unit of DifferenceRMRP-RMRIC and RMSE

- Table 5 and 6: a) I think the first line needs to be the mean and not the standard deviations; b) explain in the caption what ABS average means; c) misconfigured, please correct.

- Line 251: Figure 2. Bland-Altman plots. Bland-Altman plots between differences and mean predicted-measured resting metabolic rate using the selected equations and the new predictive equations for men and women. The dotted lines represent 1.96 SDs from the mean (limits of agreement).( I didn't understand this text, it seems that all graphs have data from the new equation. I believe it is enough to delete this expression: “...and the new predictive equations”. Also, what would dark and green colors be? please explain)

- I think it necessary to insert in the discussion some comment regarding the linear relationship used in the new formula. In the present study, better coefficients for linear relationship were verified, however, other studies have highlighted allometric relationships in RMR (Bowes HM, Burdon CA, Taylor NAS. The scaling of human basal and resting metabolic rates. Eur J Appl Physiol. 2021 Jan;121(1):193-208.)

- Reference 20 and 22 are the same. Please correct and adjust the number of citations throughout the text (sequence of numbers)

Minor questions

- Line 17: Weightin

- Line 18: + 260 RMR (point or comma after 260)

- Line 19: 43 RMR males (point or comma after 43)

- Line 21: + 43 The (point)

- spaces are missing in abstract equations

- Line 22: groupsin

- Graphical Abstract: Please, fix parentheses and font size (small)

- Line 33: Basal metabolic rate (BMR) ; (space)

- Line 81: REE, BMR, RMRPredictiveEquations: space

- Line 85: andremain

- Line 87: maleshad

- Table 1: extra spaces and missing spaces (review formatting and fix)

- Table 1: H-B Rev.by Rosa &Shizgal in kcal/d (1984): review formula formatting

- Table 1: Pavlidou (Proposed New Equations), in kcal/d (2022): this equation has the same parameters as the upper equations (Wt, Ht, Age, Gender). I suggest not separating, but I understand if the authors want to keep it that way

- Line 91: Roza and Shizgal (what is it?)

- Line 100: (Table 1).These (Please, review the manuscript for missing spaces, excess space, formatting, for example: R2 and etc.)

- Reference: a) title written in capital letters. The other references are in lower case (need to standardize (See reference 13; 14, 18, 2, 21,24, 27, 29, 31, 39, 40, 42); b) title of the manuscript is not abbreviated, while the others are abbreviated. Need to standardize (See Reference 28)

- Reference 38: missing information: editor etc

Author Response

Dear reviewer,

We are grateful for your insightful comments on our paper. We have been able to incorporate changes to reflect most of the suggestions provided. We have highlighted the changes within the manuscript. We also attach a word file with the replies to your inputs.

Happy new year.

Kind regards

Round 2

Reviewer 2 Report

no further comments

Author Response

Dear Reviewer

We sincerely appreciate all your valuable comments and suggestions, which helped us in improving the quality of the manuscript.

Thank you very much for your effort.

Reviewer 3 Report

Major questions

- Table 5 and 6: A suggestion: wouldn't it be better to join table 5 and 6 by putting the formulas on the lines?

- Reviewer: - I think it necessary to insert in the discussion some comment regarding the linear relationship used in the new formula. In the present study, better coefficients for linear relationship were verified, however, other studies have highlighted allometric relationships in RMR (Bowes HM, Burdon CA, Taylor NAS. The scaling of human basal and resting metabolic rates. Eur J Appl Physiol. 2021 Jan;121(1):193-208.)

Authors: It has been added

Reviewer: insertion in the discussion does not seem to me to be enough. I miss a little more robust discussion on the subject.

Minor questions

- Reference 31: manuscript data is missing

- Reference 48 and 44 are the same. Need to correct citation in text too

Author Response

Dear reviewer,

Thank you for your constructive comments concerning our manuscript.

We have studied your comments carefully and made the corrections which we hope to meet with your approval. We answer your questions or comments in the following texts.

- Table 5 and 6: A suggestion: wouldn't it be better to join table 5 and 6 by putting the formulas on the lines?

Answer: They've been merged

- Reviewer: - I think it necessary to insert in the discussion some comment regarding the linear relationship used in the new formula. In the present study, better coefficients for linear relationship were verified, however, other studies have highlighted allometric relationships in RMR (Bowes HM, Burdon CA, Taylor NAS. The scaling of human basal and resting metabolic rates. Eur J Appl Physiol. 2021 Jan;121(1):193-208.)

Authors: It has been added

Reviewer: insertion in the discussion does not seem to me to be enough. I miss a little more robust discussion on the subject.

 Answer: It has been added

Minor questions

- Reference 31: manuscript data is missing

Answer: It has been updated/ replaced

- Reference 48 and 44 are the same. Need to correct citation in text too

Answer: It has been corrected

Round 3

Reviewer 3 Report

Major questions

- Table 5: please review and update the title and legend of the tables after the changes made previously

Minor questions

- References: Please, separate the year from the name of the manuscripts from the references (all references).

Author Response

Dear Reviewer

We sincerely appreciate all your valuable comments and suggestions, which helped us in improving the quality of the manuscript.

Thank you very much for your effort.

Major questions

- Table 5: please review and update the title and legend of the tables after the changes made previously

 Answer: It has been corrected

Minor questions

- References: Please, separate the year from the name of the manuscripts from the references (all references).

Answer: They have been separated